# Mapping the basement of the Cerdanya Basin (Eastern Pyrenees) using seismic ambient noise.

Jordi Díaz[1], Sergi Ventosa[1], Martin Schimmel[1], Mario Ruiz[1], Albert Macau[2], Anna Gabàs[2], David Martí[1], Özgenç Akin[1,3] and Jaume Vergés[1]

[1]GeoSciences Barcelona, Geo3Bcn, CSIC c/ Solé Sabarís sn, 08028 Barcelona, Spain

[2]Institut Cartogràfic i Geològic de Catalunya, Barcelona, Spain

[3]Karadeniz Technical University, Trabzon, Turkey

*Correspondence to*: Jordi Díaz (jdiaz@geo3bcn.csic.es)

**Abstract.** Ambient seismic noise acquired in the Cerdanya Basin (Eastern Pyrenees) is used to assess the capability of different methodologies to map the geometry of a small-scale sedimentary basin. We present results based on a 1-year long broad-band deployment covering a large part of the Eastern Pyrenees and a 2-months long high-density deployment covering the basin with interstation distances around 1.5 km. The explored techniques include autocorrelations, ambient noise Rayleigh wave tomography, horizontal to vertical spectra ratio, and band-pass filtered ambient noise amplitude mapping. The basement depth estimations retrieved from each of these approaches, based on independent datasets and different implicit assumptions, are consistent, showing that the deeper part of the basin is located in its central part, reaching depths of 600-700 m close to the Têt Fault trace bounding the Cerdanya Basin to the NE. The overall consistency between the results from all the methodologies provides constraints to our basement depth estimation, although significant differences arise in some areas. The results show also that when high-density seismic data are available, HVSR and ambient noise amplitude analysis in a selected frequency band are useful tools to quickly map the sedimentary 3D geometry. Beside this methodological aspect, our results help to improve the geological characterization of the Cerdanya Basin and will provide further constraints to refine the seismic risk maps of an area of relevant touristic and economic activity.

## 1 Introduction

The objective of this contribution is to evaluate the potential of several methodologies based on the analysis of the seismic signals recorded in the absence of earthquake-generated waves, such as autocorrelations, Horizontal to Vertical Spectral Ratio (HVSR), Ambient Noise Tomography (ANT) or noise amplitude maps to define the geometry of the Cerdanya Basin (CB), a relatively small Neogene sedimentary basin located in the eastern part of the Pyrenees Axial Zone (Figure 1). The basin extends 35 km along its longer axis, has a maximum width of 5-7 km and is crossed by the Segre River, one of the main tributaries of the Ebro River. The mean altitude of the CB is 1100 m, with surrounding mountain ranges reaching 2500-2900 m.

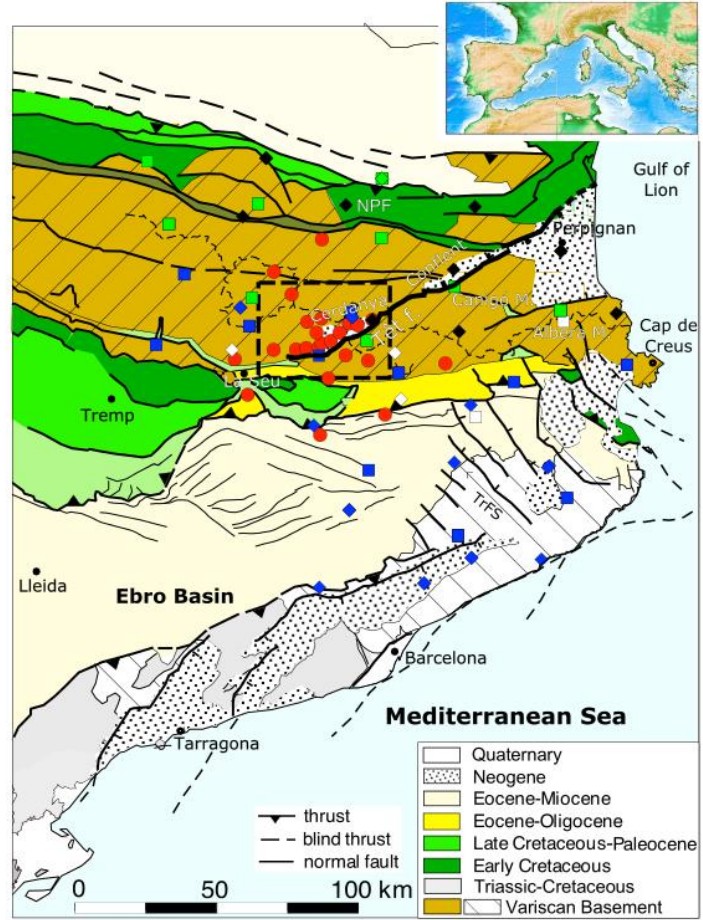

34

**Figure 1**: **Simplified tectonic map of the Eastern Pyrenees**, adapted from Vergés et al. (2019) including the main Pyrenean thrusts and Neogene extensional faults. TrFS stands for Transverse Fault System, NPF for North Pyrenean Fault. Thick black line shows the location of the Têt Fault, as outlined in Milesi et al. (2022). Red dots show the deployment of the SANIMS broad-band stations. Permanent broad-band (squares) and accelerometric station (diamonds) are included for reference. Blue: CA network; White: ES network; Green: FR network; Black: RA network. Dashed square shows the location of the map in Figure 2.

## 1.1 Geological setting

The Pyrenees, extending from the Mediterranean Sea to the Cantabrian Mountains, were built by the inversion of Mesozoic sedimentary basins and the stacking of northern Iberian crust thrust sheets to build the Axial Zone, the central part of the chain (e.g. Muñoz, 1992; Teixell, 1998). The northward underthrusting of the Iberian plate under a thinner European plate resulted in crustal thicknesses reaching 40-45 km beneath the central part of the chain (e.g. Diaz et al., 2016). However, different geophysical results have shown that the Pyrenean range does not have cylindrical symmetry (Chevrot et al., 2018) and that the eastern termination of the Pyrenees is marked by the abrupt thinning of the crust, decreasing from more than 40 km beneath the Cerdanya Basin to values close to 25 km beneath the Mediterranean shore (Gallart et al., 1980; Diaz et al., 2018). This thinning has been associated to the presence of widely distributed faults (e.g. Calvet et al., 2021; Taillefer et al., 2021), whose origin has been related to the initiation of the European Cenozoic Rifting System (e.g. Angrand and Mouthereau, 2021) or the back-arc extension leading to the opening of the Gulf of Lion (e.g. Séranne et al., 2021). The most prominent of

the normal faults in the eastern Pyrenees is the Têt Fault, extending from the coastline to the Segre valley, in the
south of Andorra, along approximately 100 kilometers. The present-day activity of the fault is still under debate,
as current displacements are low or nonexistent (e.g. Lacan and Ortuño, 2012), but some authors relate triangular
facets of the Têt fault escarpment to its recent activity (e.g. Briais et al., 1990; Calvet, 1999). The fault is divided
in two main segments; to the east, the Conflent segment, extending from the coastline to the village of Mont-
Louis, and to the west, the Cerdanya segment, extending from this point to the town of Seu d'Urgell (Fig. 1). The
present day seismic activity around the fault is minor to moderate, with most of the recorded events having local
magnitudes below 4. However, the Têt Fault could have been on the origin of the large, destructive earthquakes
in the XV century (e.g. Briais et al., 1990). From thermochronological studies along the Têt Fault (Milesi et al.,
2022), it has been observed that the most pronounced cooling of the Canigó and Carançà massifs, in the southern
footwall of the Têt Fault, occurred during the Oligocene-lower Miocene between 26 and 19 Ma, while the South
Mérens Massif in its hanging wall of the fault was not exhumed. Later on, during the Serravallian-Tortonian
between 12 and 9 Ma, the Carançà Massif shows a new cooling event, while the Canigó Massif remained
unaltered. Therefore, this segment of the Têt Fault has played a major role in the extensional evolution of the area,
that, accordingly to Milesi et al. (2022), started during the late Priabonian, in the same time than the European
Cenozoic Rifting System affecting western Europe. Since this episode, the Têt fault activity appear mainly
controlled by the opening of the Gulf of Lion.

The Cerdanya Basin is a half-graben about 30 km long developed in the NW side of the southern segment of the
Têt Fault and can be divided in two main sections located to the east and to the west of 1.85°, near the town of
Riu de Cerdanya (Figure 2). This geometry is clearly related to the position of the Têt Fault, which has a general
NE-SW trend, but abruptly changes its trend towards an E-W direction at its SW termination (Calvet et al., 2022).

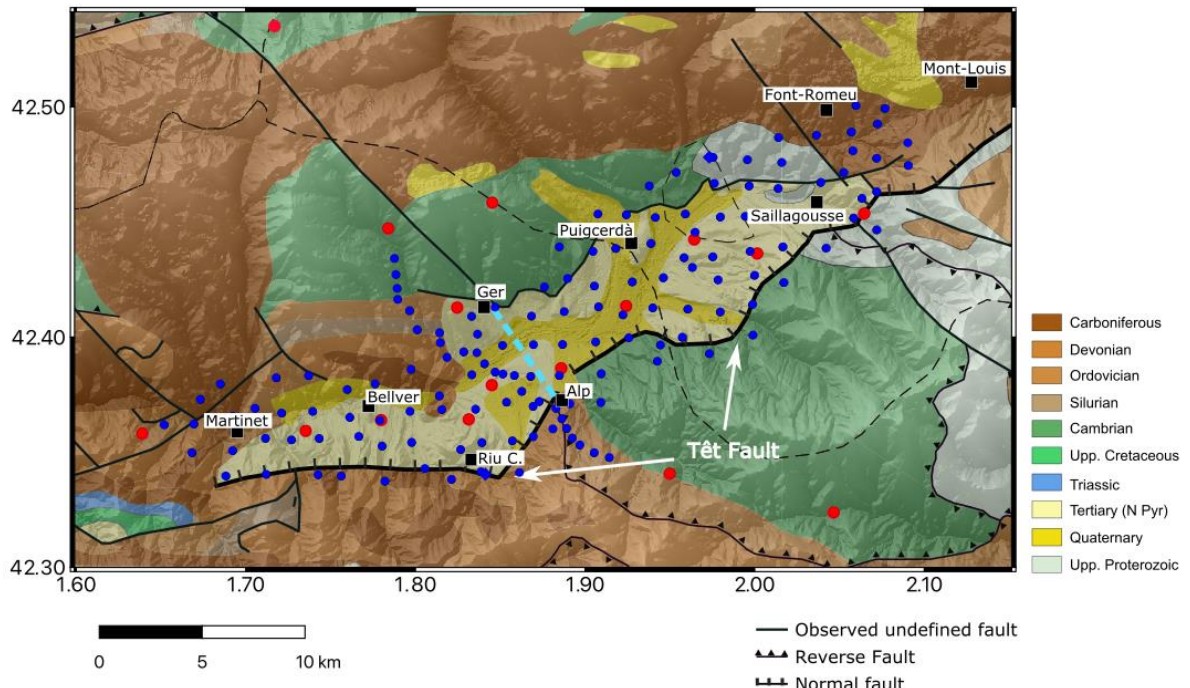


 **Figure 2: High density deployment of seismic nodes (blue dots) between April and June 2021**. Red dots show the location

of the previous broad-band deployment. Light blue dashed line shows the location of the profile presented by Gabàs et al
(2016). Black squares and labels show the location of towns. The background shows the geological map around the Cerdanya
Basin (Instituto Geológico y Minero de España and Bureau de Recherches Géologiques et Minières, 2009).

The Cerdanya basin Neogene infill deposited directly on the Hercynian basement, formed by Cambro-Ordovician
schists and Hercynian granitoids and including the Carançà and Canigó massifs to the SE and the Mérens massif
to the NW. This Neogene infill is composed of alluvial and fluvial deposits (muds, sandstones and conglomerates)
and lacustrine deposits (diatomites and thin lignite beds) with variable thickness between 400 and 1000 m, sourced
from the two sides of the basin (Roca, 1996; Cabrera et al., 1988). Two stratigraphic units, separated by a slight
discordance, form the filling of the basin (Roca and Santanach, 1996; Agustí et al., 2006). The lower unit is dated
as Vallesian and Turolian, between 11 and 5.5 Ma, while the upper one is of latest Miocene-Pliocene age between
6.5 and 6 Ma. The Neogene strata thicken and dip towards the Têt Fault and thus showing their growth pattern
(Chapter 13 in Calvet el al., 2022). The Quaternary deposits (last 2.58 million years) cover a large area of the
Cerdanya Basin and are mainly alluvial and fluvial terraces with thicknesses between a few meters and a few tens
of meters (Turu et al., 2023). Thin remnants of moraines and associated fluvioglacial terraces are found at the
confluence of the Segre and its tributary Querol River near Puigcerdà city. These Quaternary deposits mostly
extend in the NE-SW segment of the Cerdanya Basin, while they are reduced in the E-W trending southern sector
of the basin. In this sector, these deposits seem to be entrenched near the basin-basement contact, possibly
triggered by uplift and high dissection of the Neogene basin infilling that crops out to the south near the E-W
trending fault zone.

**1.2 Previous knowledge on the Cerdanya Basin geometry**
Previous geological and geophysical studies have provided information on the structure of the subsoil in the first
hundred meters depth in the CB, using vertical electric sounding (Pous et al., 1986), seismic (Macau et al., 2006)
or gravimetric (Rivero et al., 2002) methods and geological data including structural mapping, relative chronology
of the fault slickensides and depositional analysis (Cabrera et al., 1988). The most relevant contribution to the
knowledge of the basin geometry was published by Gabàs et al. (2016) and included the joint use of
magnetotelluric and passive seismic data along a high-density 2D profile across the basin, between the villages of
Ger and Alp (Figure 2). The obtained models show an average value of the electrical resistivity overburden close
to 40 Ohm·m and can be correlated with Quaternary and Neogene deposits. The derived bedrock profile has a
maximum sediment thickness of 500 m near its SE termination and an asymmetric geometry, with a smooth
increase in depth to the NW and a more abrupt change in the SE termination. This layer is a resistive zone with
electrical resistivity values between 1000 Ohm·m and 3000 Ohm·m and could be correlated with the top of the
Palaeozoic rocks constituting the basement (limestones and slates) (Roca, 1996; IGME and BRGM 2009).

**1.3 Data used**
We use the seismic data acquired in the framework of the SANIMS project (Spanish M. of Science, Innovation
and Universities, Ref.: RTI2018-095594-B-I00), which includes two different deployments. Firstly, we deployed
24 broad-band stations covering the CB and the surrounding areas with a twofold objective; investigating the
basin and providing data for regional-scale tomographic studies (Fig. 1). Ten of the stations were deployed along
an EW profile crossing the CB with an interstation spacing of 4-6 km. The rest of the instruments were deployed
forming an outer circle located about 35 km from the basin. These instruments were active between September
2019 and November 2020. Secondly, we deployed a high-resolution network covering the basin using 140 Rau-
Sercel nodes equipped with 3-component 10 Hz geophones and acquiring data at 250 samples per second (Fig.
2). The network had an interstation spacing of 1.5 km, covering an area of about 300 km$^2$ and was active for two
months, between April and June 2021. Additionally, a high-density node profile, crossing the basin along a NW-
SE line, was designed with an interstation spacing of 700 m. Although the two deployments were planned to be
operative during the same time period, the logistical constrains related to the COVID19 mitigation measures
delayed the high-density station deployment by one year.

**1.4 Receiver Functions results**
Before discussing the results provided by noise-based methodologies, we want to point out that a first piece of
information on which is the area with thicker sediments can be obtained from the inspection of the Receiver
Functions (RF) calculated with the main objective of mapping the bottom of the crust. The RF method uses the
P-to-S wave conversion at large velocity discontinuities to map subsurface structures, typically the Moho, and is
widely used to explore crustal structure. As the objective of this paper is not to analyze in detail the results from
this technique, the steps followed to calculate the RFs are described in the Supplementary Material S1.

Zelt and Ellis (1999) have described the effect of sedimentary basins on RFs, which include an apparent time lag
of the first peak, resulting from the delayed arrival of the P-to-S converted phase at the base of the sedimentary
layer and the presence of large reverberating phases that can overprint the arrival of the phase converted at the
Moho. Figure 3 shows the RF stack at the broad-band stations installed along the CB. It is easy to observe that
stacks corresponding to stations CN02 to CN10 show late arrivals of the direct P wave, with maximum time lags
for stations CN07 and CN08. These two sites show also large reverberations between 2 and 6 s, hence suggesting
the presence of a significant sedimentary cover in the central part of the CB. Further modeling of the RFs, out of
the scope of this contribution centered on the use of ambient noise, can provide additional information on the
properties of the basin (Yu et al., 2015).

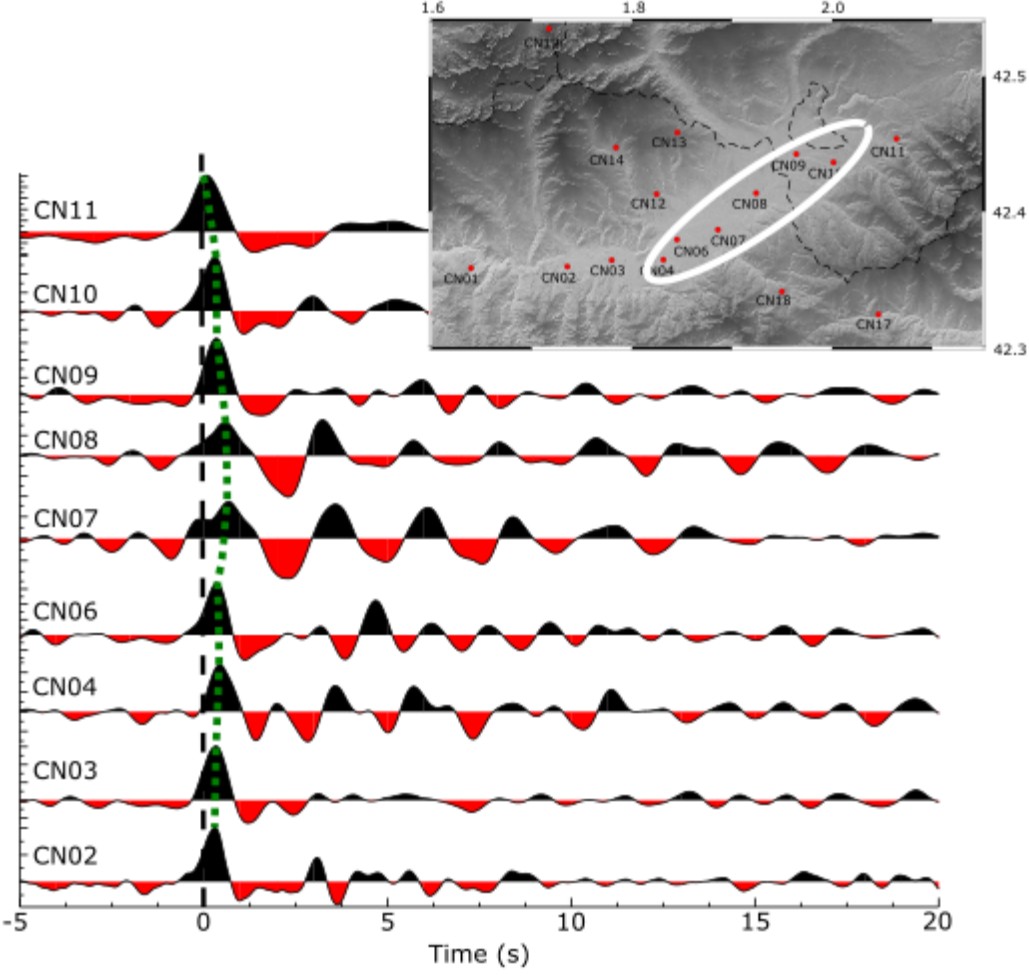

**Figure 3: Stacked RFs for the broad-band stations located along the Cerdanya Basin**. Dotted green line show the delayed arrival of the P-to-S converted phase at the base of the sediments for stations along the basin. Large reverberations are clearly observed for stations CN07 and CN08, affecting also stations CN04, CN06, CN09 and CN10. The location map in the inset shows the area with delayed RFs.

## 2 Autocorrelation methods

Autocorrelation methods are based on the evaluation of the similarity of a seismic trace with a delayed version of itself, as this similarity depends on the subsurface structure. Claerbout (1968) showed that the zero-offset Green's Function of a one-dimensional medium can be recovered from the autocorrelation of transmitted plane waves originated in the subsurface. For 2D and 3D media, Wapenaar (2004) has proved that this approach is still valid, although presence of wave fields which are not diffuse does not allow to recover the exact function. However, the obtained result, usually referred to as the empirical Green's function (EGF) to express its approximative character, is now widely used to characterize the subsurface structure.

Autocorrelation and cross correlation of ambient noise have been applied to dense station deployments to retrieve P wave reflections for crustal-scale imaging (e.g., Ruigrok et al. 2012). More recently, this approach has been used to map the Paleozoic basement in areas as the Ebro Basin (Romero and Schimmel, 2018), as it provides a fast and consistent imaging of the basement structure. However, mapping such shallow structures demands to

work in frequency bands between 1 and 25 Hz, a point that may hamper the applicability of the method due to the
dominance of local noise overprinting the weak amplitude body wave reflections. Further, the presence of
structural complexities complicates the EGFs and often results in ambiguities in the interpretation of the
autocorrelations. These ambiguities, nevertheless, can be reduced by using dense station deployments and a priori
information arising from well logs.
In this contribution we have calculated the autocorrelations for all the broad-band stations located along the CB.
We have also tried to calculate autocorrelations with the data acquired by the seismic nodes, but the quality of the
results is poor, as many resonances do appear. We think that this may be related to the high self-noise of the
geophones used by these stations that mask the low-energy reflected signals. We have tested several frequency
bands to assess the best choice for imaging the uppermost crustal discontinuities focused on this study. Finally,
the pre-processing includes the correction of the raw data to ground velocity, the band-pass filtering from 8 to 20
Hz, the division into one-hour-long non-overlapping sequences, and the rejection of those sequences containing
gaps or transient peaks. We compute autocorrelations up to a maximum lag time of 20 s using wavelet phase
cross-correlations with a complex Mexican-hat wavelet with 2 voices per octave and no decimation due to its high
temporal resolution (e.g., Addison et al., 2002). Then, we smooth the hourly autocorrelations, stacking one-day-
long consecutive cross-correlations separated by 12 hours and weighting them by the inverse of the norm between
2 and 3 seconds.
We identify the reflector associated to the base of the basin manually by selecting the first negative reflector
identified after the source reverberations having a time arrival consistent with the a priori knowledge of the area.
For most of the broad-band stations located along the CB the signal due to the selected reflectors arrive at two-
way travel times ranging between 0.4 s and 0.6 s (blue lines in Fig. 4). The Vs models obtained by Gabàs et al.
(2016) show velocities between 0.5 and 1.0 km/s in the uppermost layers. From the Vs/Vp relationship proposed
by Brocher (2005), these Vs values correspond to Vp in the range 1.75 – 2.25 km/s. Assuming Vp=2 km/s, this
results in basement depths ranging between 400 m for CN03, 640 m for station CN07 and 300 m for station CN10.
This approach provides our first quantitative estimation of sediment thicknesses in the same area where delayed
RFs have been observed. In order to assess the error of these estimations, we have calculated the depths values
obtained using Vp=1.75 km/s and Vp=2.25 km/s. The difference between these extrema cases is close to 100 m,
and the error associated to the selected velocity is therefore expected to be in the order of +/- 50 m.

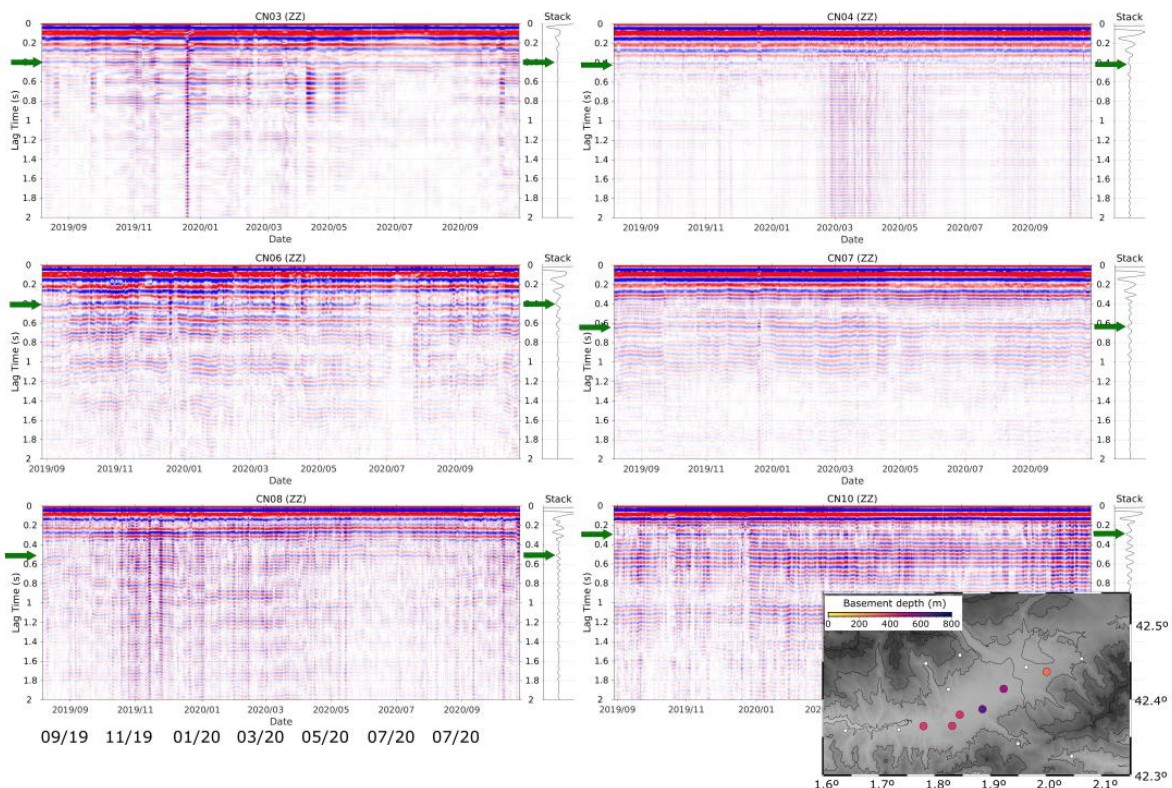

**Figure 4: Daily autocorrelograms for the vertical components of broad-band stations** CN03, CN04, CN06, CN07, CN08 and CN10, all located along the Cerdanya Basin. Dark green arrows show the reflectors interpreted as corresponding to the basement. Vertical axis refers to the two-way travel time (s). Traces are ordered by date, with the total stack shown beside each panel. The inset map shows the basement depth estimations.

## 3 Ambient noise tomography (ANT)

Ambient noise tomography is based on the extraction of the fundamental mode Rayleigh waves to measure inter-station group and phase velocity dispersion curves (e.g. Campillo and Paul, 2003; Shapiro et al., 2005; Wapenaar et al., 2010). The obtained dispersion curves are then inverted following a hybrid $l_1$-$l_2$ norm (e.g. Tarantola, 2005) criterion using the fast marching method (Rawlinson and Sambridge, 2005) on the forward problem to produce velocity maps for a set of periods.

The data gathered with both the broad-band and the nodes deployments, together with the data at the permanent stations covering the area, have been used to obtain a high resolution ANT model centered in the CB. The data processing includes correcting the raw data to ground velocity from 0.05 to 20 Hz, band-pass filtering from 0.1 to 5 Hz, decimating to 20 samples per second, dividing into one-hour-long non-overlapping sequences, and rejecting sequences containing gaps or high-amplitude signals. We compute symmetric cross-correlations up to a maximum lag time of 90 s using the wavelet phase cross-correlation and time-scale phase-weighted stack (ts-PWS, Ventosa et al., 2017)and then measure Rayleigh phase-velocity dispersion curves following Ekström et al., (2009). To estimate the average and the confidence of the phase velocity extracted from the cross-correlation ensemble per station we randomize the individual cross-correlation, subsequently stacked with the two-stage ts-

PWS, using the jackknife resampling cross-validation technique (Efron and Stein, 1981) following the resampling
strategies of Schimmel et al. (2017). Finally, we construct Rayleigh phase-velocity maps solving an inverse
problem with $l_1$-norm misfit function on the data space and a $l_2$-norm on the model space using the steepest-
descent method, and applying the fast marching method (Rawlinson and Sambridge, 2005) to solve the forward
problem.
Although the pointwise inversion to depth of the dispersion curves is still not available, the inspection of the phase
velocity maps at short periods provides a good insight on the geometry of the uppermost crustal layers. In
scenarios with strong velocity contrast such as a sedimentary basin, sensitivity kernels at short periods are highly
sensitive to the low-velocity layer. Broadly, this sensitivity increases as period reduces and the sedimentary layer
thickens in strongly non-linear manner. As the Rayleigh-wave phase velocities at periods from 1 to 2 s have their
maximum sensitivity at depths ranging between 200 – 800 m, the low velocity zones observed at the shortest
periods analyzed can be roughly interpreted as corresponding to sediments in the uppermost layer, with significant
variations in thickness along the basin. The map obtained for the shortest period available, 1.0 s, shows a clearly
defined low velocity zone covering the central part of the basin, the same area where RFs and autocorrelation
methods have already pointed to a significant sedimentary cover (Fig. 5a). The low velocity zone in ANT maps
extends to the NE following the direction of the Têt Fault, including the area near Puigcerdà, although with slightly
higher Vs values, around 2.0 km/s. To the southwest, the end of the Cerdanya Basin is delineated by velocities
around 2.2 km/s, still lower than in the surrounding areas. For periods around 1.5 s the low velocity area is similar
although the velocity contrast is lower and the southern part of the basin has only a slightly lower-than-average
velocity (Fig. 5b). For periods around 2.0 s, the presence of low velocities related to the sedimentary basin is
limited to the central area (Fig. 5c).

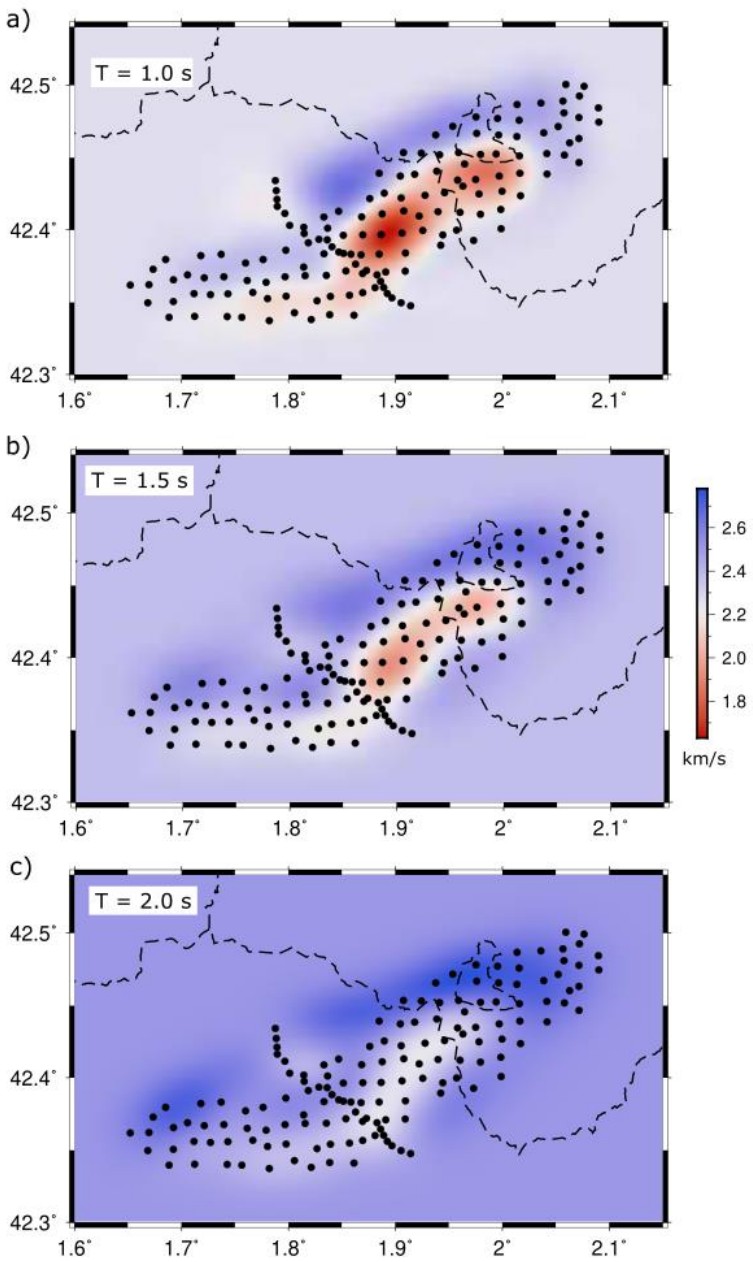

**Figure 5: Absolute Rayleigh wave phase velocity maps derived from ANT** using the high-resolution array for the shorter periods available, T=1.0 s (a), T=1.5 s (b) and T=2.0 s (c). Color scale refers to phase velocity

## 4 **Horizontal to vertical spectral ratio (HVSR)**

One of the most usual methods to characterize shallow structure using seismic data is the Horizontal to Vertical Spectral Ratio (HVSR) method (Nakamura, 1989; Bard, 2004), as it provides a reliable, fast and low-cost tool to estimate site characterizations. Analyzing the seismic background noise during different time intervals, this method allows to obtain the soil fundamental frequency ($f0$), related to the strong impedance contrast at the soil-bedrock interface (Field and Jacob, 1993). This soil fundamental frequency can then be used to estimate the depth of the soil-bedrock interface by means of empirical relations with borehole stratigraphies or velocity-depth profiles in places where this kind of results are available (Ibs-Von Seht and Wohlenberg, 1999; Benjumea et al., 2011; Akin and Sayil, 2016; Delgado et al., 2000). HVSR methods were applied by Gabàs et al. (2016) to define

the geometry of the CB along a 2D-profile, obtaining f0 values ranging between 0.3 and 1.7 Hz. The use of seismic array methods has allowed the authors to obtain shear-wave velocity-depth profiles and to then infer a scaling law between f0 and basement depths.

The processing of the data acquired by the broad-band stations and the seismic nodes to calculate the HVSR starts with correcting the instrument response to ground velocity. We filtered the data between 0.05 and 20 Hz, a frequency range wide enough considering the fundamental frequencies we can expect. Following a classical approach, we split all available data, spanning over a year for the broad-band stations and 2 months for the node deployment, into sequences of 240 s with a 50% of overlap and windowed with a Hann taper, the parametrization providing the best results after performing several tests. Similarly to Konno and Ohmachi (1998), we then smooth the spectra, applying a bank of 261 Hann filters with a quality factor (i.e., bandwidth divided by central frequency) of 20 uniformly distributed in a logarithmic scale along the above frequency range, and subsequently averaging their outputs. Finally, we tested different criteria to determine the optimum HVSR, observing that the best results were obtained when using the least square criteria, i.e., HVSR = sqrt(E{H·V}/E{V·V}) where H and V are the horizontal and vertical spectra, and the expectations E{·} is measured as the mean value after removal of outliers. Supplemental Material S2 shows some examples of the obtained HVSR.

The geophones used by the high-density array have a characteristic frequency of 10 Hz, which means that the recording sensitivity decreases for frequencies below this value. As the f0 values expected in a sedimentary basin are clearly below 10 Hz, these instruments are not the most suitable for this type of study. However, a significant number of the instruments have provided useful f0 values, even below 1.0 Hz. For the broad-band stations along the basin, the lower f0 values (0.36-0.38 Hz) are observed at stations CN07 and CN08, with values between 0.40 and 0.60 Hz for stations CN04, CN06, CN09, and CN10. Stations outside the basin, mostly located over the Paleozoic massif, do not show clear frequency peaks for frequencies below 20 Hz. Regarding the nodes, we have retained 59 valid measurements from a total of 143 (40%). A large number of the nodes installed in the central part of the basin, near broad-band stations CN06-CN08, have not provided useful HVSR measurements. We interpret that the frequency range of these instruments was not sensitive enough to the low-frequency f0 values expected for the sites located in this area. Figure 6a show the retained f0 measurements over the network.

As discussed above, Gabàs et al. (2016) have adjusted an exponential law to relate f0 and basement depth, based on their velocity/depth models. Even if this relationship was inferred for the narrow zone covered by their experiment, it still seems to be better suited for application to our case, located in the same sedimentary basin, than experimental laws published for other basins and has therefore been used to translate the new f0 measurements to basement depth estimations (Fig. 6b). Just four broad-band stations, all of them in the central part of the basin, have depths exceeding 450 m, reaching a maximum value of 530 m below station CN07. Stations located near the Puigcerdà area (1.95°, 42.42°) show depth values above 400 m, while large parts of the basin show basement depths ranging between 350 and 450 m. As expected, the thinner values are found at the locations close to the borders of the basin.

We have interpolated a gridded surface using the nearest neighbor algorithm included in the GMT software
package (Wessel et al., 2013), using a search radius of 2 minutes of arc and requiring 3 out of 8 sectors providing
data. These values have been selected by trial-and-error in order to avoid artifacts related to the interpolation and
to keep a good spatial resolution. Although this surface must be taken with caution, as the interpolation is based
on a strongly uneven point distribution, it provides our first quantitative map of the basement.

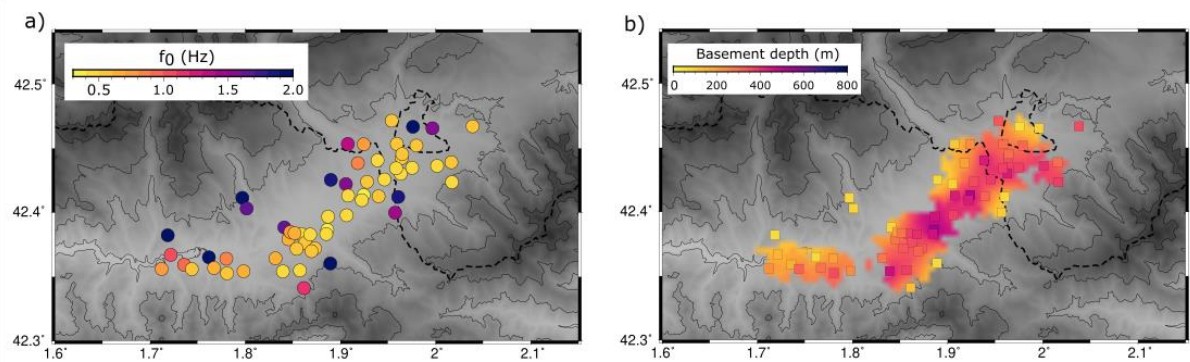

**Figure 6: Results from HVSR.** a) f0 values retrieved from broad-band stations and seismic nodes. b) Basement depths
estimated from the f0 values using the scaling law proposed by Gabàs et al. (2016). The background shows the topography
and the dashed line corresponds to the Spanish-French border.

**5 Seismic amplitude mapping**
Between 0.1 and 1 Hz, in the frequency range commonly known as the microseismic peak, the origin of the ground
vibration is mainly related to the interaction of oceanic waves (e.g. Díaz, 2016). This explains the great similarity
of the spectrograms in this range for all the analyzed stations. Background seismic vibrations at frequencies above
2 Hz in stations located near populated areas are dominated by human activities. The seismic signals show
typically a large daytime/nighttime variation, with large amplitudes during working hours and much smaller ones
during nighttime and weekends. This point has been evidenced during the recent COVID19 lockdown, when
seismic data in the 2-20 Hz has been used as a proxy of human activity, both at local scale (e.g. Diaz et al. 2021;
Maciel et al. 2021) or at global scale (Lecocq et al., 2020a). The frequency range between 1 and 10 Hz, located
between the microseismic peak and the band dominated by anthropogenic noise, provides the best opportunity to
explore the eventual relationship between seismic amplification and geological structure. Other processes, as
rainfall or wind bursts can contribute to the observed amplitudes (Diaz et al., 2023), but their effect tend to be
limited in time, while the amplification effects due to sediments should be observed continuously.

In order to analyze the amplitude variations as a function of time, the instrumental response is removed following
standard procedures. The Power Spectra Density (PSD) is then calculated to quantify the energy levels at each
frequency, using an Obspy implementation (Krischer et al., 2015) of the classical PQLX ("IRIS- PASSCAL
Quick Look eXtended") software (Mcnamara et al., 2009), based on the open-access "SeismoRMS" software
package (Lecocq et al., 2020b). The data is divided into 30-minute windows with 50% of overlap and the PSD of
each window is computed using the Welch method. The spectrograms retrieved from the PSD analysis show the
power of the seismic acceleration, expressed in decibels (dB) referred to 1 $m^2/s^4$/Hz. The inspection of these
spectrograms (Fig. 7) confirms that the day/night variations typically related to human activity dominates the

spectra at frequencies above 10-15 Hz. It can also be observed that, for frequencies above 40 Hz, episodes of increased amplitudes can be recognized at many of the stations. Recently, Diaz et al. (2023) have shown that the seismic signals at this frequency range are dominated by rainfall episodes and proposed to use seismic data as a proxy of rainfall. These observations confirm that the 1-10 Hz band is the best choice to analyze a possible relationship with the subsoil geology. The effect of anthropogenic noise is still visible in this band, in particular for nodes located close to villages or main roads, but, as shown in Fig. 7, its energy is much lower than for frequencies above 10 Hz.

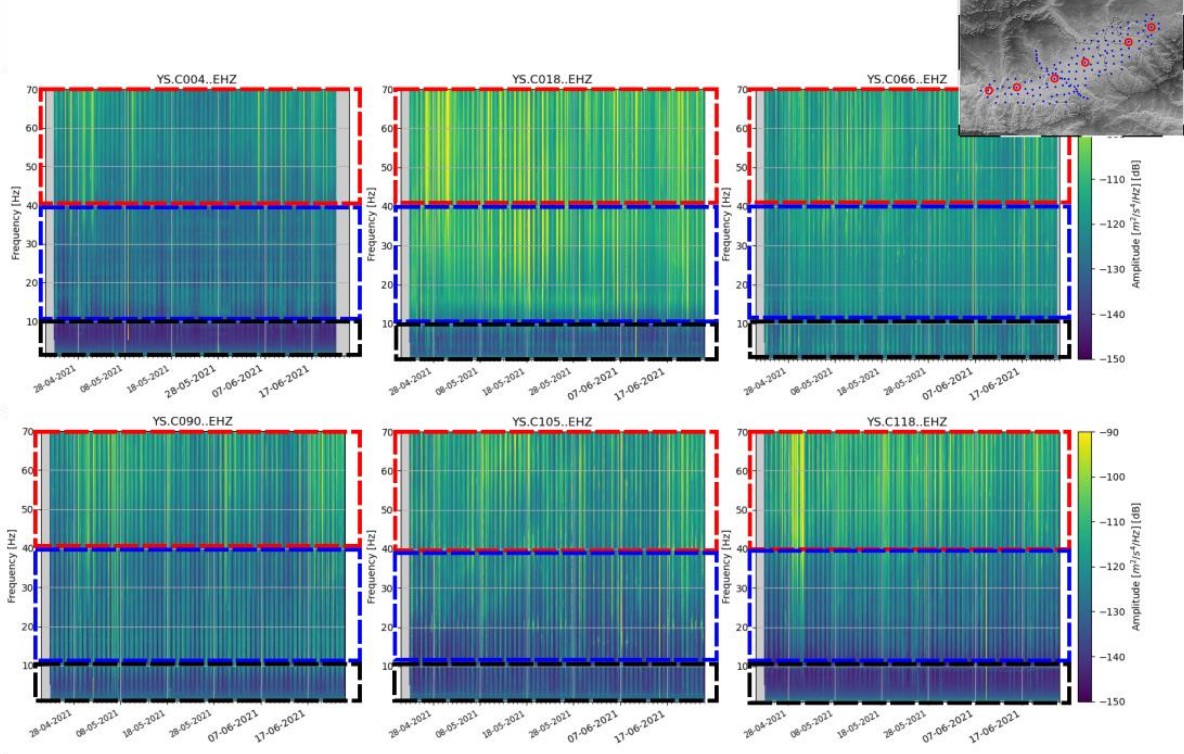

**Figure 7**: **Spectrograms for stations distributed along the basin.** Red and blue boxes show the frequency bands dominated by meteorologic and anthropogenic sources. Black boxes outline the frequency band related to site amplification.

From the calculated PSD spectrograms, we extract the median value of the power spectra in the 1-10 Hz band for the whole available records (12 months for the broad-band stations, 2 months for the nodes). The amplitude of the power spectra is calculated at intervals of 30 minutes and expressed in dB relative to 1 $m^2/s^4$/Hz. In a first stage, this procedure is applied to the broad-band stations installed along the basin (Fig. 8a). As observed, the largest values are found in the thickened area identified by RFs, autocorrelation and HVSR.

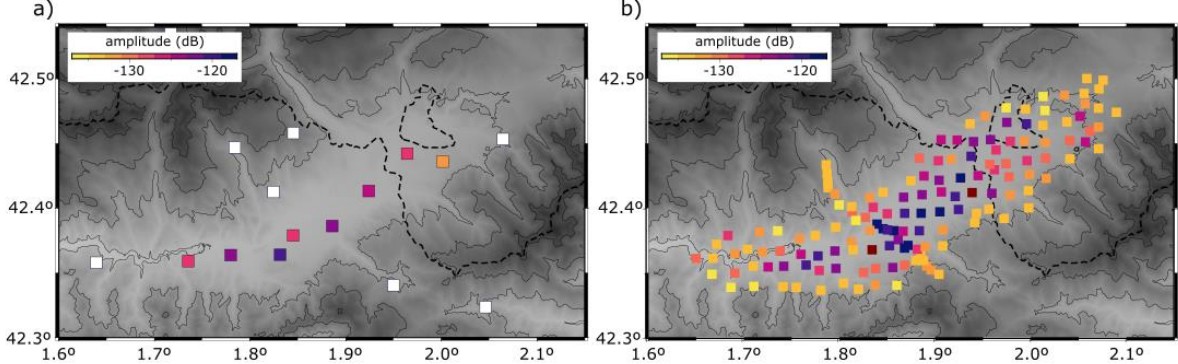

**Figure 8: Seismic power amplitude for the BB stations (a) and for the dense nodal deployment (b).** The color palette represents the median amplitude in the 1-10 Hz band, measured in dB. White squares are for broad-band stations with lower median amplitudes (out-of-scale). The background shows the topography and the dashed line corresponds to the Spanish-French border.

The same approach has then been applied to the dense seismic network available in the Cerdanya Basin. Fig. 8b shows the median values for all the nodes, clearly showing a distribution with low-level amplitude sites around the borders of the basin and large amplitude sites in the center of the basin, over the same areas previously identified as showing the largest sedimentary thicknesses.

A visual comparison between the seismic noise amplitude values and the basement depths estimated by Gabàs et al. (2016) along the Ger-Alp profile, suggest that it is possible to obtain a scaling law between both datasets (Supplementary Material S3), as there is an agreement in the relative variations along the profile of the estimated basement depths and the new seismic noise amplitude values, with the larger depths and the higher amplitudes located in the central part of the basin. Although a linear relationship can provide a general good adjustment, the better results are obtained using a degree two polynomial adjustment, following the expression:

$$depth = 3.97 * dB^2 + 1062.89 * dB + 71118.22$$

Supplementary Material 3b shows the quality of the adjustment. It can be observed that for depths exceeding 400 m, the power amplitude increases very slowly, suggesting that there is a threshold effect in the relationship between sediment thickness and amplitude amplification. In order to avoid extrapolation effects, we have limited the application of this law to noise values ranging between -133 and -120 dB.

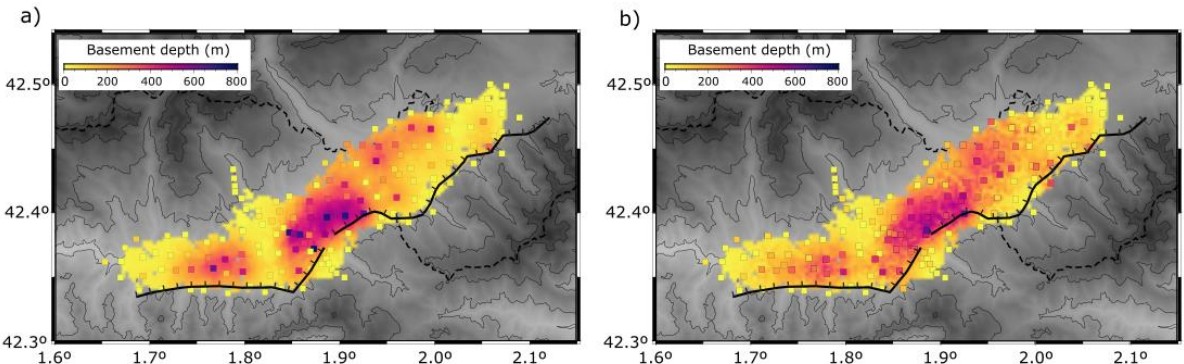

**Figure 9: Basement depths inferred from seismic amplitudes in the 1-10 Hz band** (a) and including also the HVSR-derived values (b). Thick black line shows the location of the Têt Fault. The background shows the topography and the dashed line corresponds to the Spain-France border.

Following this law, power amplitude values are converted to basement depth estimations and represented in a map (Fig. 9a). As the results derived from the nodal network have a high-density distribution, with a site located every 1.5 km approximately, it is possible to interpolate a continuous grid covering the area. As for the HVSR case, we have used the nearest neighbor algorithm included in the GMT package (Wessel et al., 2013), to obtain a continuous grid, using a search radius of 3 km, an interval of 0.3 km and requiring 6 of 8 sectors with data.

Finally, in order to check the consistency of our results, we have interpolated, using the same parametrization, a new grid using as input the basement depth estimations arising from autocorrelations, HVSR and seismic noise amplitude analysis (Fig. 9b). As observed, both models are very similar, with the deeper values, reaching values exceeding 600 m, located in the central part of the basin and a secondary maximum with depths around 300 m in the western part of the basin. The interpolation grid of all the datasets allows to obtain an averaged basement depth estimation, which is considered our final result

## 6 Discussion and conclusion

Ambient seismic noise data acquired in the Cerdanya Basin has been used to assess the suitability of different methodologies to investigate the 3D geometry of this sedimentary basin located in the Eastern Pyrenees. Autocorrelation relies on the reflection of body waves of unconstrained origin, ambient noise tomography is based on the propagation of surface waves of unknown origin between the receivers, HVSR considers the horizontal and vertical components and provides measurements in terms of frequency and seismic amplitude provides measurements in terms of energy. Therefore, all approaches are complementary, reduce ambiguities and provide a more complete picture of the Cerdanya Basin geometry. The new results provide a 3D regional scale map of the depth of the basement beneath the Neogene-Quaternary sedimentary deposits, clearly improving the knowledge on the depth of the CB, limited till now to the narrow profile analyzed by Gabàs et al. (2016) (Supplementary Material S4).

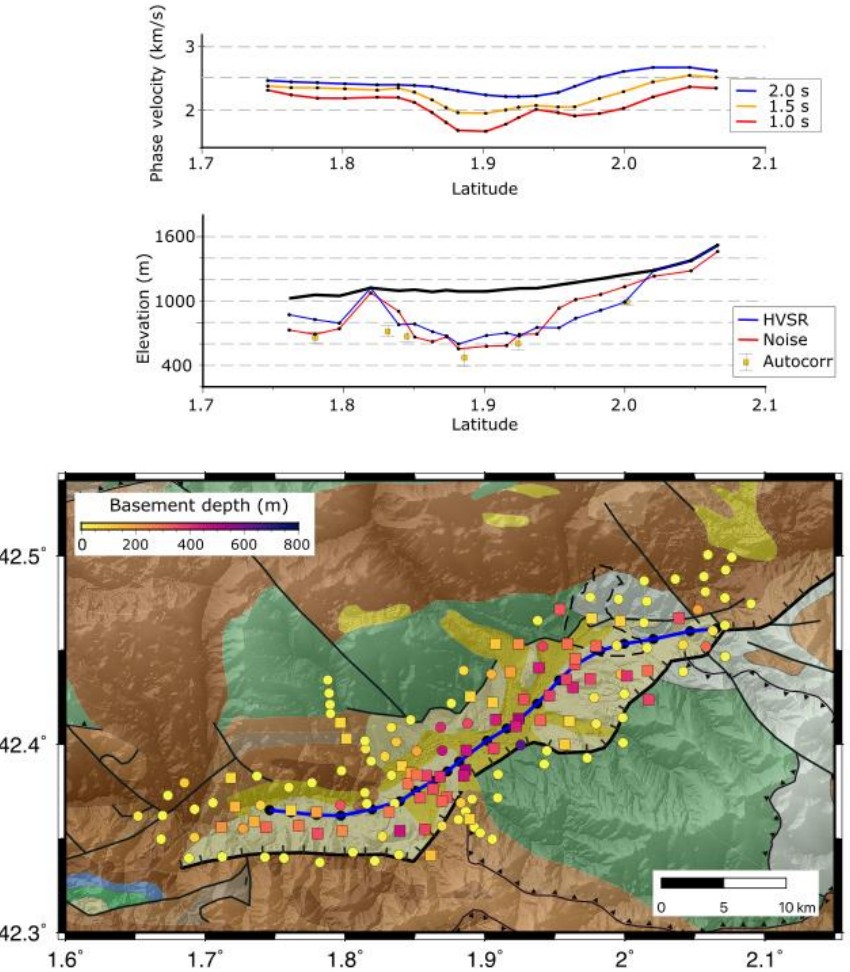


**Figure 10: Sedimentary thickness along the Cerdanya Basin**. Upper panel: Ambient noise tomography phase velocities for
different periods along the profile shown by a blue line in the lower panel. X-axis are longitudes and black dots correspond to
the points used to extract the data. Middle panel: Topographic elevation (thick black line) and thicknesses of the sedimentary
basin along the same profile. Red line shows the seismic noise amplitude estimations, blue line the HVSR estimation and
yellow squares show the depth estimations from autocorrelations. In the latter case, the estimated error (see text) is represented
by a bar. Lower panel: Location of the extracted profile overprinting the geological map (Instituto Geológico y Minero de
España and Bureau de Recherches Géologiques et Minières, 2009)

The analysis of the autocorrelations of the broad-band stations within the basin has shown that reflectors
associated to the sediment/basement discontinuity can be identified in most of these sites. Reflections in
autocorrelations need to be calibrated using boreholes for their correct interpretations. Unfortunately, no borehole
data is available in the Cerdanya Basin, but using realistic Vs values, we have estimated that the vertical
sedimentary thickness ranges between 420 and 670 m, which are consistent with the maximum preserved
Miocene-Quaternary sedimentary infill accumulative thickness of almost 800 m (Cabrera et al., 1988). A more
quantitative result arises from the use of HVSR to the broad-band stations and the high-frequency geophones
deployed with short interstation distances in the basin. Using the empirical formula proposed by Gabàs et al.,
(2016), the f0 measurements have been translated to depths. The high frequency geophones used in the node
deployment, with a natural frequency cut-off of 10 Hz, do not allow us to recover the f0 frequencies in the deeper
part of the basin, but provide interesting values for the thinner parts, hence providing a first 3D vision of the basin

geometry. The results clearly evidence that the thicker sedimentary successions are those already identified from the autocorrelation analysis, although the obtained thicknesses are lower. Finally, the analysis of the seismic noise amplitude in the 1-10 Hz band, can be interpreted as an excellent proxy of the sediment thickness along the basin. Passing from amplitudes measured in dB to basement depth is possible by correlating the noise measures at the locations studied by Gabàs et al. (2016) with the corresponding basement depths. The polynomial correlation obtained allows us to determine a 3D map of the basin, which is consistent with the HVSR and autocorrelation estimation. Additional confirmation of the results arises from the ambient noise tomography obtained using the high-density dataset. Although results for the depth inversion of this dataset are not yet available, the velocity maps at short periods, sensitive to the uppermost parts of the crust, show low velocity areas clearly consistent with the results obtained from the rest of methodologies.

It is difficult to provide a quantitative evaluation of the consistency of the different approaches, as most of the sites do not provide simultaneous measurements of autocorrelations, f0 and seismic noise amplitude. Although we have noted that there is an overall similarity in the results arising from the different methodologies, it is also clear that relevant variations do occur, in particular for some specific areas. In order to get an insight of the order of magnitude of these variations, we have represented in Fig. 10 the basement depths estimations from autocorrelations, HVSR and seismic noise amplitude along a profile crossing the whole basin. In addition, we have represented the velocity variations along the same profile in the ANT models for T=1.0, 1.5 and 2.0 s, sampling the uppermost part of the crust. The figure includes a geological map with the basement depths estimations derived from all the methodologies represented by squares colored using the same palette than for the grid in Fig. 9b. The western section of the profile shows basement depths around 300m, with the HVSR estimations thinner than those from seismic noise and autocorrelations. Near 1.8ºE, in the zone where the CB orientation changes, the basin seems to vanish, to reach its thicker section immediately to the NE. Between 1.8ºE and 1.9ºE the agreement of the results is high, with differences in the 50-100 m range. Further north, between 1.9º E and 2.0ºE, although all the results show a NE directed thinning, the differences are much larger, with the HVSR grid varying from 400 m to 250 m and the seismic noise amplitude estimations passing from 200 m to just around 50 m. The only autocorrelation estimation available in this area is close to the HVSR values. Near the NE termination of the profile, the results are again consistent. In general, the basement depths estimations derived from the seismic noise amplitude in areas with relatively low amplitude are underestimated. As discussed above, the law to pass from amplitude dB to basement depths was derived using the data published by Gabàs et al. (2016). As the amplitudes in the area covered by their profile are higher than in other parts of our region of interest, it is possible that the relationship between amplitudes and basement depth needs to be revised for the sites with low amplification (Supplementary Material S4). Further data will be needed to verify this point. Regarding the ANT results, all the profiles show a clear thickening in the central part of the basin and a gradual thinning towards the terminations of the profile. Although a more accurate comparison will only be possible once this dataset has been inverted to depth, the relative variations observed in these profiles are consistent with the basement depth estimations from the other methods. The velocity variations are larger for T=1.0 s, the period with a sensitivity kernel closer to the surface.

The differences between the results from each methodology are attributed to the different hypotheses used in each case, from the choice of a certain frequency band for the seismic noise amplitude, the formulas used to convert amplitudes of f0 values to depths or the velocities used to pass from autocorrelation TWT to depths. However, we want to highlight that the order of magnitude and the relative thickness variations derived from all the methodologies are consistent, proving that these approaches, quicker to obtain than tomography inversions, are a good option to assess the geometry of sedimentary basins.

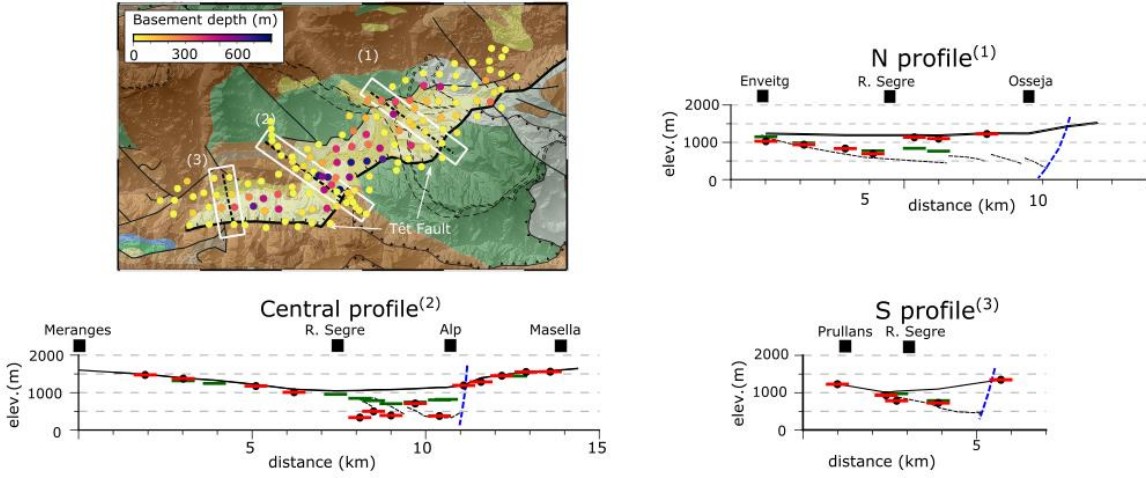

**Figure 11: Profiles along the three main domains across the basin**. X-axis is the distance along the profile and Y-axis shows the topographic elevation (black lines) and the depth of the bottom of the sedimentary basin inferred from seismic noise amplitude (red dashes) and HVSR (green dashes). The location of the profiles and the depth estimations at the sites along the profiles are shown overprinting the accompanying geological map (Instituto Geológico y Minero de España and Bureau de Recherches Géologiques et Minières, 2009). Black dashed lines represent the base of the sedimentary basin in the models by Calvet et al. 2022. Blue dashed lines show the location of the Têt Fault, projected from the geological cross-section by Calvet et al. (2022).

Figure 11 shows the basement depth transect along the high-density node profile crossing the central domain of the Cerdanya Basin, altogether with two profiles crossing its northern and southern domains, following the geological models shown in Calvet et al. (2022). The northern profile shows the base of the basin displaying a gentle deepening towards the SE with its deepest part located in its central part close to the town of Puigcerdà. This deepest part of the basin nearly coincides with the limit between the areas covered by alluvial fans of granitic and non-granitic sources. As reflected in the geological maps (Instituto Geológico y Minero de España and Bureau de Recherches Géologiques et Minières, 2009), the western part is covered by alluvial materials of granitic origin transported from the north. To the east, the basin is covered by alluvial fans from slate or limestone source reaching the basin from the ENE (Cabrera et al., 1988). The seismic noise amplitude in this eastern zone are low, resulting in basement depths of just $100 – 50$ m, clearly below the geological models in Calvet et al (2022). The results from HVSR and autocorrelations are scarce in this area, but tend to provide thicker depth estimations. The basement depth estimations derived from ambient noise seems to be underestimated in this area of relatively low amplitude, either because the used law to pass from dB to depths does not work properly for low amplifications or because the alluvial materials have a different seismic amplification than more consolidated sediments. Further

geological and seismic studies will be needed to fix this point. The central profile shows a sedimentary thickness reaching 700 m and a steep margin in the location of the Têt Fault limiting the basin along its SE side. Although the results from seismic noise amplitude and HVSR show some differences, they are generally consistent with the en-echelon geometry proposed for the central profile in Calvet et al. (2022) south of the Segre River. However, our results suggest that the sedimentary basin reaches the Segre River, a point not documented in the geological profile. The seismic noise amplitude and the HVSR results clearly depicts the location of the Têt Fault. The profile crossing the westernmost part of the CB along a north-south direction shows a basin thickening to the south and a step discontinuity beneath the trace of the Têt Fault, an image consistent with previous models based on geological observations (Cabrera et al., 1988; Calvet et al., 2022).

The conclusion of this study is that the analysis of ambient noise and HVSR in dense (large-N) seismic networks is a useful tool to obtain information on the 3D geometry of sedimentary basins. Although the obtained estimations must be taken with caution, as significant differences between the estimations derived from both methods can appear, in particular for areas with low amplification, the overall models provide a simple procedure to estimate the uppermost crustal structures. Ambient noise tomography is expected to provide better constrained results, but the processing and inversion requirements are clearly higher and the inversion of the dispersion curves to depth benefits from a previous knowledge of the crustal structure, a point that can be provided by the previously described methods. Besides, autocorrelation methods provide an independent procedure to check the consistency of the results and can provide useful constraints, in particular if well log data are available.

The results on the geometry of the Cerdanya Basin derived from this study are expected to provide additional constrains to better understand the role of the Têt Fault in defining the geometry of the Cerdanya Basin and its present-day degree of tectonic activity. On the other hand, our results can contribute to refine the seismic risk maps in this area with important touristic activity, as a good knowledge of sedimentary basins is a key point to estimate the seismic vulnerability.

**Competing interest**

The authors declare that they have no conflict of interest.

**Acknowledgments**

We acknowledge the Geo3Bcn-CSIC LabSis Laboratory (http://labsis.geo3bcn.csic.es) for making their seismic stations available for this experiment. We want to thank the local authorities that help to find appropriate and secure locations for the instruments.

This work has benefited from open source initiatives such as Obspy (Krischer et al., 2015), SeismoRMS (Lecocq et al., 2020b) and GMT (Wessel et al., 2013). The codes implementing WPCC and two-stage ts-PWS (Ventosa et al., 2017, 2019) are open source under the LGPL v3 license and available at https://github.com/sergiventosa.

This is a contribution from the SANIMS project (RTI2018-095594-B-I00), funded by the Ministry of Science, Innovation and Universities of Spain. JV has benefited from complementary funding of the ALORBE project (PIE-CSIC-202030E10).

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
