# Peer review of "Mapping the basement of the Cerdanya Basin (Eastern 1"

_EGUsphere, 2022_

## Author Response (AR1)

**Reviewer 1 comments:**

**In the present manuscript, the authors use a variety of passive seismic methods to constrain the basement depth of the Cerdanya Basin in northeastern Spain, drawing on data from a rather sparse deployment of broadband stations as well as a much denser array of geophones. The authors apply four different approaches (autocorrelations, ambient noise tomography, HVSR, amplitude ratios), with the stated aim of exploring their usefulness for basin depth determination. While this is a good conceptual idea, and while the analysis with each of the methods seems to be well done (as far as I can tell), the manuscript is not convincing, mainly because it does not provide a detailed comparison between the results obtained with the different methods. I thus recommend major revisions, and outline my main points and suggestions below, followed by more specific comments with line numbers.**

**General comments:**
**1. The principal strength of the manuscript is that it combines four different methods that can constrain depth to basement in different ways. However, this principal strength is not exploited much at all, since no meaningful comparison and/or cross-plotting of different results is provided. Figure 9 only puts different results next to each other, but it is not easy to assess differences between methods with these images. The discussion on this is also extremely short, it is only stated a few times that the methods are consistent, without mention on what order differences between methods or between newly obtained results and those of Gabas et al (2016) actually are. A comparison with geological data is only provided for results of one method, where it would be much more interesting whether there are trade-offs, i.e. regions where one method performs better (or simply different) than another. This would also be in line with one of the central statements made in the manuscripts, that combining different methods is preferable to using only one.**

As discussed below, we have included a new figure (Fig. 10) allowing the reader to compare directly the results of each method along a profile crossing E-W the Cerdanya Basin. The discussion on this comparison has now been extended in the manuscript, providing more details on the coincidences and differences of the different methods. The former Fig. 9 has been suppressed.
Regarding the Gabas et al (2016) results, we have now added a new supplementary Figure (Suppl. Material S4) to compare their basement depth estimations with the values derived from seismic noise amplitudes and HVSRs for the same sites. As observed, the differences for most of the sites are well below 50m , although for a couple of sites the difference is larger, not far from 100m. Regarding the  comparison with geological data, the basement depth estimations used in figures 10 and 11 arise from the joint interpolation of seismic noise, HVSR and autocorrelation results. This point has now been clearly stated in the text.

**I would propose the following:**

**a) All constraints on whether sediments are present or not, either beneath single stations (HSVR, autocorrelation, RFs, amplitude ratios) or in the space between (ambient noise tomography; a velocity contour that separates sediment from bedrock in the period maps of Figure 4 could be picked) should be combined in a single Figure, and compared to geological evidence. It would, for example, be interesting to see whether any of the methods shows evidence for sediments in the region in the northeast where amplitude**

**ratio results (showing no sediments) are in conflict with geological evidence (Figure 10, N profile).**

The results from seismic noise, HVSR and autocorrelations are now shown along a profile crossing the Cerdanya Basin in the new Fig. 10. The differences between the results from different methods are now discussed in the text, with a particular attention to the zone with sediments but low seismic noise.

It is not possible to extract a velocity contour from the ANT results, as they are not inverted to depth. In order to follow the reviewer's recommendation, we have included in Figure 5 (previous Fig. 4) the velocity variations along a EW profile crossing the entire basin for periods of 1.0, 1.5 and 2.0 s. These profiles show that the velocity variations are consistent with the results of the other methods.

As better discussed now in the manuscript, the HVSR basin depth estimations in the northeast sector are larger than those provided by seismic noise amplitude. However, we want to point out that the latter dataset also identifies a sedimentary cover in this region, although it appears to be thinner than in other sectors.

**b) As no ground truth for basement depth in the form of borehole data seems to be available, the only benchmark to compare against is the short profile of Gabas et al. (2016). Instead of just plotting depth data from this profile into a map (Figure 9a), it would be much more meaningfuld to compare all obtained depth estimates with the different methods to Gabas et al. (2016) with a profile section (similar to what is shown in Figure 10 for other areas). Obtained offsets should be discussed, also taking into account that Gabas et al. (2016) mainly used HSVR.**

As commented before, we have now added a new supplementary figure showing graphically the comparison between our results and those by Gabas et al (2016) and we have now discussed this point in the manuscript.

**c) For the station-wise methods, it would be interesting to see plots of differences between basement depth estimates for the same station using different methods. Possible systematic offsets or more random differences should be quantified and discussed.**

We think that the new Figure 10, discussed above, solves this problem.

**d) The only plot that currently combines data from different methods is Figure 8b, that somehow contains HVSR and amplitude ratio data. There needs to be a clear explanation about what was done there, to me it is unclear what is shown in that figure. What was done for stations that had an HVSR and an amplitude ratio depth, were they averaged? While providing a final depth-to-basement map that combines the different methods is useful, there needs to be transparency on how it is obtained. Also: Why are only two methods combined and not all available information, and why does Figure 10 not use this depth estimate but that from the amplitude ratios?**

We have specified in the text that the depths estimations arising from seismic amplitudes, and HVSR methods were combined to interpolate a new grid. In fact, we have rebuilt now this grid incorporating also the basement depth estimations from autocorrelation, although they have a limited influence in the final result. As stated in the text, the ANT is not inverted to depth, so its results can not be integrated into the final grid.

**2. Figures are generally provided at a too low resolution, and axis labels, text in the figures etc. is nearly always hard or impossible to read at 100% size.**
Part of this problem is related to the conversion of the original figures to the png format to include them into the Word document. In any case, we have tried to increase the size of the labels for all the figures.

**3. Manuscript structure: I would recommend to add a Section on the geological setting of the Cerdanya Basin after the Introduction, where all available geological constraints on sediment thickness and provenance are gathered. At the moment, some of this information is contained in the Introduction, and some is in the Discussion. Having it all in one place would be preferable.**
As stated in the introduction, the main objective of the contribution is to evaluate the potential of several methodologies based on the analysis of the seismic noise and not to discuss in detail the geological implications of our study in the Cerdanya Basin. This is why we decided to outline the geology of the zone in the Introduction section, without entering into detailed descriptions of the geology of the zone.
However, attending the reviewer request, we have now divided the introduction in four subsections (Geological setting, Previous knowledge of the basin geometry, Data used, RF results) " and extended the description of the geological setting, with a special attention to the description of the sedimentary cover.

**Specific comments:**

**l.13: What was the approximate station spacing of the coarser deployment?**
The coarser deployment had a station spacing of 4-6 km along the Cerdanya Basin, and 14 additional instruments drawing a circle with a radius of approximately 30-40 km, as already stated in the text (l 48). We think that it is not necessary to provide this explanation in the abstract, as its length is limited. On contrary, we think that stating the interstation distance for the nodes is relevant, because such kind of deployments are still not usual.

**l.14: any reason the sequence of methods is different here compared to how they appear in the manuscript text?**
No, there was no reason for that. We have now swapped the references to ANT and HVSR in the abstract.

**l.21: There is literally nothing about seismic risk in the manuscript, so I would leave this out of the abstract**
We agree on the point that we do not discuss seismic risk in the manuscript. However, what is stated in the abstract is just that we expect that our results "will provide further constraints to refine the seismic risk maps". We do not pretend to provide a contribution on risk, but yes to provide the bases for a future development. We think that this is true and interesting for the reader, who may be interested in working on this subject.

**l.24: such as**

Corrected

**l.25/26: extends 35 km along its long axis**
Done

**l.28: delete "of"**
done

**l.34: "geological studies": reformulate; geological studies is not a way to determine basement depth**
We have now clarified that we refer to "geological data including structural mapping, relative chronology of the fault slickensides, and depositional analysis (Cabrera et al., 1988)"

**l.37: It would be nice to see this profile location in one of the maps in Figure 1 (or to be able to find the two mentioned villages)**
We have now represented this profile in Fig 1c

**l.39: sediment thickness?**
corrected

**l.46: remove "period"**
done

**l.50: corner frequency as well as sampling frequencies should be mentioned here**
we have now stated that the geophones had a 10 Hz corner frequency and that the acquisition rate was 250 sps

**l.65: centered on**
corrected

**l.84: why only the broadband stations and not the dense nodal array? The previous sentence stated that using dense arrays can reduce the ambiguities in autocorrelation results, thus I expected that the dense array was actually used...**
We have tested the use of autocorrelations with the high-density node deployment, but the quality of the results is poor, as many resonances do appear. We think that this may be related to the high self-noise of the geophones that mask the low-energy reflected signals. We have now included this explanation in the manuscript.

**l.87: "complex Mexican-hat wavelet"; should be either explained or a reference given**
"Mexican-hat" is the name of a wavelet type commonly used in this kind of processing, due to its high temporal resolution. We have now included a reference to document this point, and modified the text to: "... a complex Mexican-hat wavelet with 2 voices per octave and no decimation due to its high temporal resolution (e.g., Addison et al. 2002)...."

**ll.91/92: it should be mentioned how the picks shown in Figure 3 were obtained, i.e. what was the criterion? For some of the stations, I see something like a weak phase where the pick is indicated (hard to see due to low-resolution figure and small labels...), for others (e.g. CN10) I don't see why the arrow is where it is**

The picks in Figure 3 were obtained manually, taking the criteria of selecting the first negative (blue) reflector identified after the source reverberations having a time arrival consistent with the a priori knowledge of the area. Due to the reflection coefficient at the base of the sedimentary layer, the amplitude of the reflection must be negative. As stated in the text, we agree on that the autocorrelations provide weak information if no further information is available. However, with the help of complementary data they can provide additional key constraints to the basement characterization.

**l.94: remove "Click or tap here to enter text"**
done

**l.95: how is it consistent? RFs were only used to infer where sediments should be present or absent, so this correlates with where autocorrelations show a phase for the bottom of the sediments? This should be pointed out on a map then.**
We have reworked the sentence to clarify that the stations with autocorrelations providing evidences of the presence of sediments (shown in the inset of Fig 3) are coincident with those showing anomalous RFs.

**l.103: together**
Corrected

**l.105: If the larger model is not used in the present study, why mention it here?**
We have suppressed the reference to the two ANT models:
"The data gathered with both the broad-band and the nodes deployments, together with the data at the permanent stations covering the area, have been used to obtain a high-resolution ANT model centered in the CB."

**l.110: some quick explanation what a jackknife sample means in the present context would be good**
Jacknife resampling is a well-known cross-validation technique used in statistical studies. We have now reworked the paragraph to clarify this point: "We compute symmetric cross-correlations up to a maximum lag time of 90 s using the wavelet phase cross-correlation and time-scale phase-weighted stack (ts-PWS, Ventosa et al. 2017) and then measure Rayleigh phase-velocity dispersion curves following (Ekström et al. 2009). To estimate the average and the confidence of the phase velocity extracted from the cross-correlation ensemble per station we randomize the individual cross-correlations, subsequently stacked with the two-stage ts-PWS, using the jackknife resampling cross-validation technique (Efron and Stein 1981) following the resampling strategies of Schimmel et al. (2017)."

**l.111: what is PWS (abbreviation was not introduced)?**
We have now stated that PWS means phase-weighted stack (see previous comment).

**l.115: how many period maps were produced, and at what periods? Why was no pointwise inversion of dispersion curves for a 3D S-wave velocity model attempted?**
We produced 20 Rayleigh phase and group velocity maps at periods ranging between 1.03 and 6.9644 s. We are working in the calculation of pointwise inversion of the dispersion curves, that we plan to present to the scientific community in a paper focused in ANT.

**l.116: what was the rationale for the choice of these periods (1.035, 1.414 s)?**

In the revised version we present the 1.035, 1.517 and 2.07 periods in order to provide a good sampling of the results obtained at short period, in the range most sensitive to the depth range of the expected sedimentary basin thickness. The velocity variation for these periods along a transversal profile is presented now in Figure 10.

As commented before, we plan to publish the complete ANT results in a new contribution that should be submitted in the next months.

**l.146: this should have been mentioned in the beginning, where the instrumentation was described**
We have now mentioned the corner frequency of the geophones in the Introduction.

**l.191: what were the sampling frequencies of the stations? Figure 6 shows spectra up to 70 Hz, so at least 150 Hz?**
The nodes were acquiring data at 250 sps. This is now stated in the Data subsection of the Introduction.

**l.203: It would make a lot of sense to plot a map where inferences of sediment presence/absence between different methods are summarized and compared**
In fact, Figures 6 (HVSR), and Fig 8 (Seismic noise) already show the zones were sediments are inferred. We have modified the color scale to make clearer the differences in depth estimations. The new Figures 10 and 11 allow to compare these results with the geology.

**l.224: How the map in Figure 8b was compiled is unclear to me. Both Fig. 8a and 8b are interpolations over station-wise depth estimates, so how does Figure 8b combine HSVR and noise amplitude results for single stations? Is the average between the two single values taken, or something else? Needs explanation, and plotting something like differences between HSVR-derived depths and noise amplitude derived depths at stations that have both would be quite interesting (see General Comments)**
We have now reworked the paragraph to explain that what we do is to interpolate a new grid using as inputs the depth estimations from seismic noise and HVSR. In the case of having two different estimations in the same location, we just leave the algorithm to work with both values. In practice, this is equivalent to use the mean value. As commented above, in the second case we have now included the autocorrelation estimations, although if they provide only few points and do not modify substantially the result.

**ll.230-260: this is almost exclusively repetition of previously mentioned results**
We think it is useful to the reader to retaken in the Discussion / Conclusions section the main results of our study.

**l.252: by how much are they lower, and why is that the case? This needs more discussion than one half-sentence**
The differences between the results derived from different methods is now discussed in more detail in the Discussion and Conclusions section, commenting that even if there is an overall consistency in the results, in some locations the differences can be large, in the order of 200m. We have modified our conclusions to state that our approach is a good tool to monitor sedimentary basin geometry, but that more accurate data is needed to get precise estimations.

**ll.261ff: It would have made more sense to have a short section on what is known about the geology of the Cerdanya Basin at the beginning of the manuscript, instead of dropping bits and pieces here**

We have moved this paragraph to the new subsection 1.3 "geological setting", were the geology is discussed in more detail.

**l.271: What did Calvet et al (2022) base their geometry on?**
The geological cross-sections presented in Fig. 13.4 by Calvet et al (2022) are constructed based on the thickness and dip of the successive Neogene strata filling in La Cerdanya Basin. Field observations are reliable in the distal region of the basin (away from the fault) where the oldest sediments are placed discordantly on the Paleozoic basement. The deep geometry of La Cerdanya Basin close to the fault trace is more difficult to observe but the inclinations of more than 20 degrees of the younger strata towards the fault indicate a typical half graben geometry as depicted in the cross-section (at least for cross-sections A and C). Cross-section B shows a syncline in the Neogene deposits close to the fault trace indicating different kinematics of the fault or tectonic complexity in the hanging wall of the fault and thus more difficult to interpret.

**l.276: is it only the amplitude ratio method that does not see sediments in this region, or is this across all tried methods? Needs more detailed evaluation!**
Both the seismic amplitude and the HVSR depth estimations result in reduced sedimentary thicknesses in the eastern part of the basin, although the HVSR estimations are larger than those from seismic noise. This is now explained in the manuscript and we suggest that the origin may be related to a poor adjustment of the amplitude/depth law for low amplifications, due to the fact that the Gabas et al (2016) data used to calibrate the law are located in an area of relatively high amplitudes.

**l.289: This claimed consistency has not really been shown (see General Comments). Also, you cannot claim that the different methods are complementary and consistent at the same time, it can only be one or the other**
We are convinced that the coincidences and differences between the results derived from the different seismic methodologies have now been better discussed, probing that there is an overall consistency between the different results.

**l.291: Why especially there?**
The reference had to do with the difficulty of direct geological observations in these types of environments, but was not justified and has now been removed.

**Figures:**

**Figure 1: subfigures b and c are of low quality, and it is hard to read labels and text. The same is true for white labels in subfigure a. Station names at least for the broadband stations and the nodal ones for which results are shown in other figures (spectra in Figure 6) should be supplied in the maps.**
The original version of the subfigures are vectorial PS/PDF files and its resolution should be fine. The problem is probably related to the conversion in the Word document.
In order to make the information in the figure more readable, we have now split the figure. The new Figure 1 shows the broad-band seismic stations used overprinting a simplified tectonic map, in which we have highlighted the location of the Têt Fault. The new Figure 2 shows the node deployment overprinting a detailed geological map of the Cerdanya Basin and surrounding areas, recovered from the IGME Geological map of the Pyrenees and including the corresponding legend.

**Figure 2: It is nearly impossible to see which station is which, the inset map is small and the station names in it would only be legible at 300% magnification, if the resolution were not so low**

We have now increased the size of the inset and increased the font size used to show the station names in the new Fig. 3. Resolution should be fine in the original figure.

**Figure 4: Same color scale should be used for the two maps, so that velocities can be compared; Caption says that Rayleigh wave phase velocities are shown, but color bar is labeled Vs.**

We have now modified the figure (now Fig. 5) to include 3 periods (1, 1.5 and 2 s), representing the velocity variations in the uppermost crust using the same palette, although this lead to a poorer definition of the velocity variations for each particular case.
We have also corrected the label to use the appropriate term, "Phase vel".

**Figure 5: The caption mentions Gabas et al. (2014), which is not in the reference list...or is Gabas et al. (2016) meant?**

We refer to Gabas et al 2016; The caption has been corrected.

**Figures 6/7: It is confusing that the color scales are basically reversed between figures, with yellow colors showing highest amplitudes in Figure 6 and lowest amplitudes in Figure 7. In Figure 6, I do not know which station is which in the inset (needs labels).**

Color scales used for both figures represent different parameters. The spectrograms in the new Fig 7 represent the seismic amplitude for all frequencies, expressed in dB. The new Fig. 8 shows the mean power amplitude for frequencies in the 1-10 Hz band. The color palette used in both cases are also clearly different (blue-green-yellow and yellow-orange.red.purple). We would like to keep this choice, as the Viridis color scale used for spectrograms is becoming a standard choice for this kind  and the yellow-to-purple palette allows the reader to visualize the variations in sediment depths.

**Figure 10: There needs to be a clearer picture of the trace of the Tet Fault, either here or in Figure 1a (it is shown there, but the label is really hard to find/read).**

The figure trace of the Tet fault has now been highlighted in the new version of figure 11. We have also highlighted this fault in the general map shown in Figure 1.

**Figure S2: Color scale for basement depth is reversed compared to all other figures**

This was indeed a mistake that has now been corrected. Suppl. Figure 2 uses now the same color convention than the rest of the figures.

**Reviewer 2 comments:**

**Dear Authors and Editor,**

**Thanks for the opportunity to review the manuscript egusphere-2022-1138, "Mapping the basement of the Cerdanya Basin (Eastern Pyrenees) using seismic ambient noise." By Diaz et al. 2022. The methods are solid, but there is an obvious lack of critical assessment of their results. Also, in some places, the methodology is not clear enough to be easily replicated. I would strongly urge the authors to consider writing their methods such that anyone can reproduce their results.**
**To improve the usability and applicability of the methods, additional work is required (as jotted down below). In my opinion, this manuscript is suitable for publication upon major revisions, including significant improvement in the text and figures.**

**Introduction**

**I found this section the weakest link in the paper and, in my opinion, requires significant improvement. In its current form, the Introduction fails to provide a relevant and thorough geological background (sedimentary history of the basin), state of knowledge of basement depth and geophysical properties of the Cerdanya Basin, known challenges, and, if possible, previous studies that have used the four methods to estimate basement depth in different regions (successfully or unsuccessfully). Currently, after 40 lines, the authors discuss the instrumentation, which is usually the 'Data' section.**
We have now rebuilt this section, dividing it in three separate subsections (Geological setting, Previous knowledge of the basin geometry and Data used, RF results) and extending the description of the geological setting, with a special attention to the description of the sedimentary cover.

**The authors have not provided any information on how the Receiver functions were calculated (para starting at line 55). I would recommend either making a separate section on RF as it doesn't fit in the 'Introduction' or moving this preliminary analysis to Supplemental Information.**
We think that making a separate section of RF will not be appropriate, as the paper focuses on the different uses of ambient noise records to investigate the geometry of the basement. To attend the reviewer's recommendation, we have included now as Supplementary material S1, a text describing the procedure followed to calculate the RFs.

**Quantitative assessment**

**My major concern with the manuscript is the lack of quantitative comparison between different methods. Given three of the methods give basement depth below each seismic station, it should be relatively straightforward to directly compare the values through a Figure. A subsequent discussion on the variations between the different methods and the sources thereof would be extremely useful for future users of these methods, and I believe is within the scope of this manuscript.**
As commented in the replies to Review #1, we have now included a new figure (Fig. 10) comparing the results from the different methods along a profile crossing the Cerdanya Basin. An extended discussion on this subject has also been added to the revised manuscript.

**Uncertainties**

**In its present state, the authors have not addressed the important question, "what are the sources of errors in each method?" Each method, as the authors mention, has their own inherent assumptions; thus, it's important for the reader to know how that translates to basement depth uncertainties. Without a discussion of the sources and values of uncertainty, it is difficult to acknowledge the results and the subsequent conclusions.**

As in many other cases, it is difficult to provide a quantitative value of the uncertainties for our results. Note that largest uncertainty may not arise from a measurement error but from interpretative questions as for instance which is the best frequency band to analyze noise, which is the autocorrelation reflector corresponding to the base of the basin, or which is the rule to move from the $f_0$ values in HVSR to basement depth. We think that one of the strengths of our contribution, in particular in this revised version, is to show together estimations derived from different methodologies, providing to the reader a good insight on the level of confidence of the results.

**Referencing**
**Throughout the manuscript, I found the referencing to be less than adequate. There were several instances in the manuscript which could've used a reference (e.g., line no. 80-90, 99-101, 140-145,167 etc). Especially given the four different methods used, this paper could potentially serve as a benchmark for future studies trying to apply one or all these methods for basement estimates. Thus, having thorough referencing in each section would be beneficial for the readers. Further, the papers cited seemed to be quite parochial.**

We do not agree of that the referencing is "less than adequate", even if in some particular cases a reference could be missing. Regarding the specific points raised by the reviewer:

- Line 80-90: we add a reference to document that the Mexican-hat wavelet is a common option in this kind of studies. The choices of the band-pass filtering, the window length and the weighting schema were found to be the best options during the data processing and do not derive from any reference.
- Lines 99-101: We have added general references to ANT. The use of the l1-l2 criterion is based on our direct experience
- Lines140-145: This processing schema is derived from our experience.
- Line 167: We included here a reference to the GMT package (that was already cited later).

We do not share the impression of using a "parochial" criterion for choosing the references, in particular if compared with many of the papers published by other research teams.

**Additional comments:**
- **Line 60 – 61: "..phase lag in the direct-P..". Although it is usually okay to say phase lag in the direct P, however, technically it's incorrect. The direct P still arrives at t=0s, as the RF is centred on it. The delayed peak is because of the arrival of the Psb phase, which is the P converted to S at the base of the sediments. On the radial RF, the direct P has little to no energy as it arrives almost on the vertical component due to slow velocities in the top layer. See Yu et al. (2015; JGR), Cunnigham and Lekic (2019; GJI), and Agrawal et al. (2022; GJI) for clarification. Please correct this in the Figure 2 caption as well.**
    We have reworked the text and the figure caption to clarify this point,
    "...which include an apparent time lag of the first peak, resulting from the delayed arrival of the P-to-S converted phase at the base of the sedimentary layer and the presence of large..."

**Typos in referencing: line 59, 112, 247.**
Corrected

- **Line 34-35 – "However, the results obtained were rather unclear and difficult to correlate with lithological information." Could you please elaborate this?**
  The discussion on the geological setting has been reworked.

- **The reference on line 66 (Yu et al. 2015) is inaccurate. Please check.**
  We have checked the reference, but we don't see nothing wrong.

- **Line 84 – 89: Please clarify if this methodology is adopted/motivated by previous studies or designed by the authors. Clarify what classifies as "high-amplitude signal".**
  We have now reworked the paragraph to "We have tested several frequency bands to assess the best choice for imaging the uppermost crustal discontinuities focused on this study. Finally, the pre-processing includes the correction of the raw data to ground velocity, the band-pass filtering from 8 to 20 Hz, the division into one-hour-long non-overlapping sequences, and the rejection of those sequences containing gaps or transient peaks".
  We think that the new redaction solves the issues pointed by the reviewer.

- **Line 91: "…related to the bottom of the basement." I would strongly suggest the authors clarify what do they mean here and throughout the manuscript by 'basement'. Is it the base of unconsolidated sedimentary rocks or the start of crust? Thus, it is important to provide information about the sedimentary history of the area in the Introduction, as suggested earlier.**
  Attending also the comments of Reviewer #1, we have now provided more information on the geology of the area and, in particular of the properties of the sedimentary infill (see section 1.1). We think that it is now clear that we refer to the bedrock.

- **Line 91 – 95: Please comment on how the two-way travel times were picked from the autocorrelograms because there is obvious subjectivity in the process. Please clarify how you obtain the Vs velocity from Gabas et al. (2016) below each station. Also, reiterating it's the P-wave two-way travel time could be useful as well.**
  The picks in Figure 3 were obtained manually, taking the criteria of selecting the first negative (blue) reflector identified after the source reverberations having a time arrival consistent with the a priori knowledge of the area. This is now stated in the text.
  Gabas et al (Fig 6) provided a 2D velocity model used to fix the velocity range to pass from TWT to depths.

- **Line 95: unintentional text.**
  Corrected

- **Line 95 – 96: "…which are consistent with the insights provided by RFs." How so?**

We have reworked the sentence to state that: "This approach provides our first quantitative estimation of sediment thicknesses in the same area where delayed RFs have been observed."

- **Line 116 – 124: Please comment on the spatial resolution of the obtained Vs models. Also, why the obtained Vs velocities in CB are much higher than (> 1.8 km/s) than what's used in Section 2 (0.5 – 1 km/s), from Gabas et al. (2016). Given the authors state that the obtained velocities are sensitive to 0.2 - 0.8 km, why have the authors not used their Vs velocities for time-to-depth conversion in Section 2? It might make more sense to start with Ambient Noise tomography as the first result as well.**
  As the ANT data has not been inverted to depth it is difficult to compare the velocities with those by Gabas et al. As stated in the text, "..the Rayleigh-wave phase velocities at periods from 1 to 2 s are highly sensitive to sediments at depths ranging between 200 – 800 m.." However, the sensitivity kernel of such phases encompasses a larger depth range. Note that the 0.5-1.0 km/s used to pass autocorrelation TWT to depths, correspond to the uppermost 400 m in Gabas et al 2016. The mean value for the upper 800 m is closer to 1.8 km/s, not far from the results of ANT.
  Regarding the suggestion of moving the ANT section, we would like to keep the proposed order, as we are convinced that it is the best option for the reader.

- **Line 128: "..soil fundamental frequency." I would recommend using the term site instead of soil.**
  Done

- **Line 137 – 144: No references in this method paragraph. Please elucidate the origin of the steps taken.**
  We have clarified now that the band-pass filter has been chosen after several test. Regarding the data slicing and tapering, this is a classical approach in HVSR studies, as we state now in the text. Regarding the use of the least-square criteria, we explain now that we have tested different criteria to decide the best result.

- **Line 158 – 160: "...than experimental laws published for other basins...". Please substantiate this claim.**
  Moving from f0 to depths is done by using experimental laws obtained for previous authors in other settings. Here we are in a better position, as the Gabas et al models allow a more accurate conversion.

- **Line 173: "..to the interaction of oceanic waves (Díaz 2016)". I'm not sure if this is the seminal reference. An 'e.g.' in front should fix it or 'see review by'.**
  Clearly this is not the seminal reference for this, but it is a contribution that discuss not only the sources of noise through the seismic spectra. We have added "e.g." to clarify this point.

- **Line 231-235: a run-on sentence?**
  We have simplified the sentence to "Autocorrelation relies on the reflection of body waves of unconstrained origin, ambient noise tomography is based in the propagation of surface waves of unknown origin between the receivers, HVSR consider the horizontal and vertical components and provides measurements in terms of frequency

and seismic amplitude is based on vertical data and provides measurements in terms of energy."

- **Line 238, 250 etc: I'm not entirely convinced by the authors use of the word '3D' to describe their basement depth results. I'm struggling to see how extrapolating multiple point-data on a surface could be classified as rendering a "3D vision of the basin geometry".**
  What we mean is that we have a 3D image of the sedimentary geometry (basement depth estimated in a large number of surface points).

- **Line 249 – 250: "..are not able to recover the f0 frequencies in the deeper part of the basin, but provide interesting values for the thinner parts, hence providing a first 3D vision of the basin geometry". What do the authors mean by "interesting"? I'm afraid this line makes little sense in any case.**
  We have changed "interesting" by "useful", but we are convinced that the sentence has sense, as it states that using high frequency geophones is a valid option in this environment.

- **Line 256 – 257: "..which is fully consistent with the HVSR and autocorrelation estimation". How? This is where a quantitative comparison figure would be useful to substantiate such sentences.**
  We have suppress "fully". The comparison between the results of the different methods has now been extended.

- **Line 260: Please clarify what does 'previous results' mean.**
  We have reworked the sentence to "..consistent with the results obtained from the rest of methodologies".

- **Line 262: "The CB can be divided in two main sections, the eastern NE-SW trending section and the E-W trending western part (Fig. 9f)". I fail to see these two sections. I highly recommend the authors modify the manuscript such that it makes sense for a reader not acquainted with the area.**
  This paragraph has now been moved to the Introduction and reworked to make it easier to follow for the readers not familiar with the region.

- **ICGC 2016: Instead of referencing the geological map, it would be perhaps more suitable to cite the geological papers. It seems inadequate to cite a geological map.**
  We do not think that citing a geological map is an inadequate procedure, as they compile a lot of scientific information

- **Line 274 – 277: Since the legend is not provided for the geological map, I'm not sure where the alluvial fan is. Nonetheless, it seems improbable that despite the presence of young sediments, none of the four methods suggests the presence of sediments. Perhaps the authors can consult the geological studies in the region to find how thick the sediments could be in this region? It would be worthwhile, in my opinion, to resolve/address this as there seems to be enough coverage in the easter part (if I'm not mistaken).**

We have reworked this section to state that the seismic methods show a thinner sedimentary cover in the eastern part of the basin than for the available geological cross-sections. The new geological map presented in Fig 2 includes a legend caption

- **Line 289 – 299: The conclusion paragraph needs significant improvement. I don't mean to be harsh, however, it is written in a colloquial style and rather hastily. I would recommend the authors rethink their main results and write a cohesive and impactful conclusion.**
  The Discussion and Conclusions section has been reworked

**Figures**

**Font size for the lat long and legend could be increased in most figures. It's hardly legible.**
We have now increased the readability of the figures

**Figure 1: I found Figure 1 to be rather confusing. Firstly, the legend in 1a is incomplete, while none is provided for 1c. Figure 1c is also used later in Figure 9 and 10, thus, without a legend, the colours are meaningless. There also seems to be lot of text on the geological map, which could easily be removed for clarity.**
We have divided the former figure 1 in two different figures. The new figure 1 shows the general geological setting and the location of the BB network, while the new Figure 2 show a detailed geologic map of the Cerdanya Basin, including the corresponding legend, together with the location of the large-N deployment.

**Figure 3: This needs significant improvement on various fronts. Subfigures are not aligned properly and are unevenly spaced. The inset figure is too small to see, the legend more so. I'd recommend curtailing the y-axis to 1 sec on the correlograms. Clarify in the caption that it's the vertical component and two-way travel time of P wave.**
The technical details of the figure have been modified and the caption states now that they are vertical components and that the time scale refers to two-way travel time.
Regarding the y-axis extension, we prefer do not modify it, to provide the reader a better overview of the whole dataset.

**Figure 5 and 8: Incomplete captions.**
We have verified the captions of the renewed figures to assure that they provide all the relevant information.

**Figure 9: This figure is mostly redundant. Except for a and f, all subfigures have already been in previous figures. Moreover, in f, it's hard to see anything. Instead of this Figure, a comparative Figure, as suggested earlier, would be useful.**
We agree on this point. Former Fig. 9 has been suppressed, including now a new figure directly comparing the results of the different methods.

**Figure 10: Please annotate on the map the three profiles.**
We have now annotated the profiles using numbers.

---

## Author Response (AR2)

**Editorial board:**

**1 Title page does not follow the journal standards. The names of the authors should not be shortened; section "Corresponding author" is missing. Please edit it according to the information provided by SE: https://www.solid-earth.net/submission.html#manuscriptcomposition /Title page**

We have included the full names of the authors, included the "Corresponding author" information and changed the format of all the manuscript to the Copernicus template

**2. Please remove the supplementary material from *.pdf manuscript since it should be placed only in supplement.**

We have removed the supplemental material from the manuscript

**Reviewer #1**

**The authors have significantly modified the manuscript and extended the Introduction as well as the Discussion part. The new or modified figures and text parts now provide an improved Introduction to the local and regional geology as well as an in-depth comparison of the results that were obtained with the different approaches.**

**All the larger points I had raised in my review were resolved with these changes, and I believe that a lot of value has been added in the revised manuscript. I still found a few minor issues while reading, which I will list below. I thus recommend minor revisions, noting that resolving these comments should not take much time.**

**l. 28: the word "analysis" seems to be in the wrong place here**
Corrected

**l. 47: the word "stations" should be added after "accelerometric"**
Done

**ll.56-58: Maybe add a reference for the crustal thickness? not cylindrical should read non-cylindrical**
We have added a reference to Diaz et al 2016, where a crustal thickness map from controlled source profiles and RFs is presented. Attending also the Reviewer #2 comments, we hace reworked the second sentence to: "However, different geophysical results have shown that the Pyrenean range does not have cylindrical symmetry (Chevrot et al., 2018) and that the eastern termination of the Pyrenees is marked by the abrupt thinning of the crust, decreasing from more than 40 km beneath the Cerdanya Basin to values close to 25 km beneath the Mediterranean shore..."

**l. 79: in the hanging wall**
Corrected

**General comment to Section 1.1: It would be nice if the villages mentioned in the text would be shown on the map in Figure 2, so that the reader can follow what locations are being discussed**
We have now included in Figure 2 the location of the main towns and other geographycal references in the text

**l. 257: citation should be without the bracket**
Corrected

**l. 263: descent**

Corrected

**l.308ff: I don't understand this sentence...should maybe be split into two.**

As a result of a cut-and-paste problem during the previous revision, the sentence included a couple of words that made it difficult to understand. We have now rephrased to: "Following a classical approach, we split all available data, spanning over a year for the broad-band stations and 2 months for the node deployment,  into sequences of 240 s with a 50% of overlap and windowed with a Hann taper, the parametrization providing the best results after performing several tests."

**l.312: "their" refers to what here?**

This was just a mistake. We have now corrected to "Similarly to Konno and Ohmachi (1998), we then smooth the spectra, applying..."

**l. 468: should be Gabas et al. (2016), not 2014**

Corrected

**lll.486/487: should be reformulated**

We have reformulated the sentence to ".. which are consistent with..."

**l.492: do instead of does**

Corrected

**l. 506, 525, 573, Caption Figure 10: in these places (and maybe elsewhere), results from noise amplitudes are referred to as "from seismic noise", which is unclear since all methods used seismic noise. Amplitudes should be mentioned in all these places, to make clear which method is meant**

We agree on this observation and we have now moved from "seismic noise" to "seismic noise amplitude" along the manuscript (l. 451, 475, 514, 525, 579, 583)

**Figure 2: In the legend, it should say reverse fault (not inverse)**

Corrected

**Figure 5: The color bar label should be km/s (not m/s); in the Caption, the word "absolute" shows up twice, and T=2.0 s is subfigure c (not b)**

Corrected

**Reviewer #2**

**Dear authors,**

**I appreciate your efforts to improve the manuscript and address the reviewer's comments. I only have the following minor comments, which should be quick to address.**
**Please note the line numbers are from the track changes manuscript.**

**line 85-86: grammar mistake**

Attending also the Reviewer #1 comments, we have reworked the sentence to: "However, different geophysical results have shown that the Pyrenean range does not have cylindrical symmetry (Chevrot et al., 2018) and that the eastern termination of the Pyrenees is marked by the abrupt thinning of the crust, decreasing from more than 40 km beneath the Cerdanya Basin to values close to 25 km beneath the Mediterranean shore..."

**line 94: rephrase "along approximately 100km". Grammatically incorrect.**

We have rephrased to "..to the Segre valley, in the south of Andorra, along approximately 100 kilometers". We think that this is grammatically correct.

**line 101: below 4 what? MLv, M?. Please provide unit.**

We have now stated that we refer to local magnitudes. The specific typ of magnitude do not seem to be relevant here.

**line 102:"could be on the origin." this doesn't make sense. Please edit the phrase.**

We have changed to "...However, the Têt Fault could have been on the origin of the large, destructive"

**line 108-111: This sentence needs reframing. Its rather convoluted now.**

We would prefer to keep the sentence, as we think it is clear enough

**line 117-118: multiple grammatical mistakes.**

We have reworked the sentence to "....which has a general NE-SW trend, but abruptly changes its trend towards an E-W direction at its SW termination"

**line 213: referencing Yu et al. (2015) for modelling sedimentary layers using ambient noise method is not correct. This was pointed out to the authors in the first revision, and they said: "We have checked the reference, but we don't see nothing wrong".**

**Sorry to be a pedant; this is inaccurate. Yu et al. (2015) developed a method to reduce/suppress the sediment reverberation energy in the receiver functions. The resonance filter was built using the auto-correlation properties of the receiver functions. Perhaps the authors are mistaken and looking at the wrong article. Yu et al. (2015) don't use ambient noise method.**

In fact, our text we were not stating that Yu et al refer to ambient noise: "Further modeling, out of the scope of this contribution centered on the use of ambient noise, can provide additional information on the properties of the basin (Yu et al., 2015). "

In order to clarify the sentence and avoid misinterpretations, we have changed the sentence to: "Further modeling of the RFs, out of the scope of this contribution centered on the use of ambient noise, can provide additional information on the properties of the basin (Yu et al., 2015)."

**Figure 1:**

**Legend is still incomplete. What do the light grey and dark grey colors mean? Please provide a reference for the outline of Tet fault. All the geological data used should be referenced, or the source website clearly mentioned.**

The legend has now been completed, explaining that the grey areas correspond to Triassic-Cretaceous domains out of the Pyrenees. We have also included a reference to the Milesi et al 2022 to document the outline of the Têt Fault. As already stated in the caption, the map is based on Verges et al. 2019.

**Figure 3:**

**Authors have missed editing the Caption 3 (re: "..phase lag in the direct-P..".).**

We have changed "delayed direct P-wave time lag" to "delayed arrival of the P-to-S converted phase at the base of the sediments" in the Figure 3 caption

**Figure 8:**

**Please increase the font size of the legend, as reviewer 1 pointed out during the first revision.**

Done

**Figure 10:**

**please annotate x-axis as longitude in the Figure.**

Done